



# 1 How useful are gridded water resources reanalysis and evapotranspiration

## 2 products for assessing water security in ungauged basins?

Elias Nkiaka[1], Robert G. Bryant[1], Joshua Ntajal[2,3], Eliezer I. Biao[4]
[1]Department of Geography, University of Sheffield, Sheffield, S10 2TN, UK
[2]Department of Geography, University of Bonn, 53115 Bonn, Germany
[3]Center for Development Research, University of Bonn, 53113 Bonn, Germany
[4]Laboratory of Applied Hydrology, University of Abomey-Calavi (UAC), Cotonou, Benin
Elias Nkiaka (Corresponding author): e.nkiaka@sheffield.ac.uk
Postal Address: Department of Geography, University of Sheffield, Sheffield, S10 2TN, UK
**Abstract**
Achieving water security in ungauged basins is critically hindered by a lack of in situ
hydrometeorological data to assess past, current and future evolution of water resources in those
areas. To overcome this challenge, there has been a shift toward the use of freely available satellite
and reanalysis hydrometerological products. However, due to inherent bias and uncertainty, these
secondary sources require careful evaluation to ascertain their performance before being applied
in ungauged basins. The objectives of this study were to evaluate the performance of nine gridded
water resources reanalysis (WRR), and eight evapotranspiration (ET) products and to estimate
the relative uncertainties in monthly basin-scale water balance evapotranspiration (ET$_{WB}$) in eight
river basins located in Central-West Africa. Evaluation results highlight strengths and weaknesses
of the different WRR and ET products in simulating discharge dynamics and ET estimates
respectively across the basins. Analyses further revealed that the relative uncertainties in monthly
ET$_{WB}$ range from 4–25 %  with a significant increase in magnitude during the rainy season while
river discharge is the dominant source of uncertainty. Our results further revealed that the
performance of land surface models (LSMs) and global hydrological models (GHM) in
simulating river discharge is strongly influenced by the model structure, input data and spatial
resolution. Differences in ET estimates from the different ET products may be attributed to model
structure and the input data used in driving the models. Results from this study suggest that
gridded WRR and ET products are a useful source of data for assessing water security in
ungauged basins. However, given the plethora of products available, it is imperative to evaluate
their performance in representative gauged basins to identify products that can be applied in each
region.



## 1. Introduction

River discharge is one of the most important hydrological variables underpinning water resources management, aquatic ecosystems sustainability, flood prediction, and drought warnings at different scales (Mcnally et al., 2017; Couasnon et al., 2020). However, observed river discharge data is often not available at the exact location where critical water management decisions need to be made (Neal et al., 2009). This is especially the case in developing and semi arid/arid regions where hydrometeorological gauging stations are sparse (Van De Giesen et al., 2014; Krabbenhoft et al., 2022), while the number of existing stations is declining (Rodríguez et al., 2020). Despite the acute shortage in observed data, developing regions are areas that are more vulnerable to adverse hydroclimatological conditions (Byers et al., 2018; Kabuya et al., 2020). Furthermore, achieving water security in ungauged basins in developing regions remains a critical development challenge as climate change, population growth, rapid urbanization, and economic growth continue to exert pressure on available water resources under hydrological uncertainty (Flörke et al., 2018; Hirpa et al., 2019). This highlights the urgent need for more reliable data to better assess past, current, and future evolution of water resources, and to predict extreme hydroclimatological events so that better strategies can be put in place to enhance water management and mitigate the impact of extreme events (Nkiaka et al., 2020; Slater et al., 2021). Water security in this study refers to the availability of sufficient quantities of water for human use and ecosystem sustainability.

Evapotranspiration (ET) is another important hydrological variable that represents the linkage between water, energy and carbon cycles and ecosystem services and is the second largest process in the hydrological cycle after precipitation (Zhang et al., 2019). Therefore, ET plays a critical role in water availability at different scales. As such, accurate estimates of ET are also crucial for water management operations such as basin-scale water balance estimation, irrigation planning, estimating water footprint, and assessing the impact of climate change on water availability. However, globally, in situ ET monitoring stations are also scarce while the existing monitoring network cannot provide sufficient information on the temporal and spatial trends of ET at large scales (Laipelt et al., 2021). ET data scarcity may therefore limit our ability to understand changes in the hydrological cycle and water security in the context of environmental change and hydrological uncertainty.

To enhance water security in ungauged basins, there has been a progressive shift toward the use of gridded data derived from satellite and reanalysis (Odusanya et al., 2019; Nkiaka, 2022). This is because gridded data products can provide high spatial resolution and long-term homogeneous data for previously unmonitored areas at scales that are suitable for studying



changes in the hydrological cycle and for water management applications (Sheffield et al., 2018).
Several gridded data products with global coverage have been produced in recent decades.
Examples of reanalysis products include Watch Forcing Data applied to ERA-Interim (Weedon
et al., 2014) and Climate Forecast System Reanalysis (Saha et al., 2014). There is also a plethora
of satellite products for different hydrometeorological variables such as precipitation,
temperature, soil moisture, and ET. For satellite derived ET estimates, it is worth noting that this
variable cannot be directly measured by satellites, but rather derived from physical variables
observed by satellites from space such as radiation flux. As such, satellite derived ET estimates
could rather be referred to as model outputs constrained by satellite data. Considering the way
gridded ET products are derived, they tend to suffer from large biases (Weerasinghe et al., 2020;
Mcnamara et al., 2021) and therefore need to be validated before use. In fact, it is argued that
validating gridded ET products is an essential step in understanding their applicability and
usefulness in water management operations (Blatchford et al., 2020).

Previously, much attention in the development of gridded environmental data was focused

on hydrometeorological variables such as precipitation and temperature. However, rapid
advancement in computer technology has led to the development of gridded water resources
reanalysis (WRR) with quasi global coverage using both land surface models (LSMs) and Global
Hydrological Models (GHMs) driven by satellite and reanalysis data. Examples of WRR products
include the Global Land Data Assimilation System [GLDAS] (Rodell et al., 2004), "The Global
Earth Observation for Integrated Water Resources Assessment" [eartH2Observe] (Schellekens et
al., 2017), and the Global Flood Awareness System [GloFAS-ERA5] (Harrigan et al., 2020).
Several studies have demonstrated that model-based gridded WRR products can be used as an
alternative to observe river discharge in ungauged basins to: (1) understand hydrological
processes (Koukoula et al., 2020), (2) support transboundary water management (Sikder et al.,
2019), (3) identify flood events (Gründemann et al., 2018; López et al., 2020), and (4) support
national water policies (Rodríguez et al., 2020). These examples demonstrate that WRR products
have great potential for addressing water security challenges in ungauged basins. Despite their
numerous advantages, model outputs from WRR are also fraught with uncertainties resulting
from errors in the forcing data, model structure, and the parameterisation of the physical processes
in the model scheme (Koukoula et al., 2020). Therefore, it is necessary to evaluate the
performance of these products against observed river discharge where available.

Whilst the use of outputs from WRR in water management has gained significant attention

in many ungauged areas such as Asia and Latin America (López et al., 2020; Rodríguez et al.,
2020; Sikder et al., 2019), they remain largely under-utilized in Africa. For example, there are





only a few case studies reporting on the use of these products in the Upper Blue Nile River basin
(Koukoula et al., 2020; Lakew et al., 2020) and the Zambezi River basin (Gründemann et al.,
2018). Considering the scale of water insecurity in Africa -compounded by acute data scarcity
(Nkiaka et al., 2021), we feel that evaluating the performance of gridded WRR products in Africa
may enhance their adoption in water management in ungauged basins in the region. On the other
hand, several studies evaluating the performance of gridded hydrometeorological variables in
Africa have focused mostly on precipitation (Dinku et al., 2018; Satgé et al., 2020) while a few
studies that have evaluated gridded ET products focused on large basins, (Blatchford et al., 2020;
Weerasinghe et al., 2020; Mcnamara et al., 2021) and mostly adopting an annual timescale. This
may be attributed to the large scale of the basins which is ideal for the application of satellite data
and the coarse spatial resolution of some of the ET products. The availability of high spatial and
temporal resolution ET products means that it now possible to evaluate these products in small-
to medium-size basins and at a higher temporal resolution. Lastly, considering that the water
balance concept has been used widely to evaluate gridded ET products, most studies did not
account for uncertainties in basin-wide water balance evapotranspiration ($ET_{WB}$) even though
such uncertainties could be large (Baker et al., 2021). These are the key knowledge gaps that this
study will seek to address.
Focusing on eight basins of different sizes in Africa, the objectives of this paper were to:
(1) evaluate the performance of eartH2Observe Tier 1 and other WRR products in simulating
discharge in the basins, (2) evaluate the performance of eight gridded ET estimates across the
basins and (3) estimate the relative uncertainties in $ET_{WB}$ in the basins. Considering that only a
few studies have attempted to evaluate gridded WRR and ET products over Africa, this paper
contributes to the contemporary debate on the performance of these products and how there can
be used to assess water security in ungauged basins.
**2.  Materials and methods**
**2.1. Study area**
The selected basins are located in Central-West Africa ranging in size from 9,000 km$^2$ to 499,000
km$^2$ (Figure 1). Rainfall in the region is mostly controlled by the north-south movement of the
intertropical convergence zone (ITCZ). The main criteria for selecting the basins were: (1)
availability of observed river discharge data and (2) for the period of the available discharge data
to coincide with the period when gridded WRR and ET data are also available. Additionally,
some of the selected basins are facing substantial water security challenges caused by population
displacement from conflicts in the Sahel and Lake Chad regions (Kamta et al., 2021; Nagabhatla
et al., 2021). The evaluation timestep was determined by the timestep of river discharge data.





Shapefiles for all the basins were obtained from HydroSHEDS, locations of the discharge gauging
stations were obtained from the respective data sources while the area of each basin was
calculated from the basin shapefiles. HydroSHEDS drainage network offers the unique
opportunity to generate watershed boundaries for GRDC gauging stations using a proofed dataset
and applying a consistent methodology. Table 1 shows that some of the basins are transboundary
in nature.

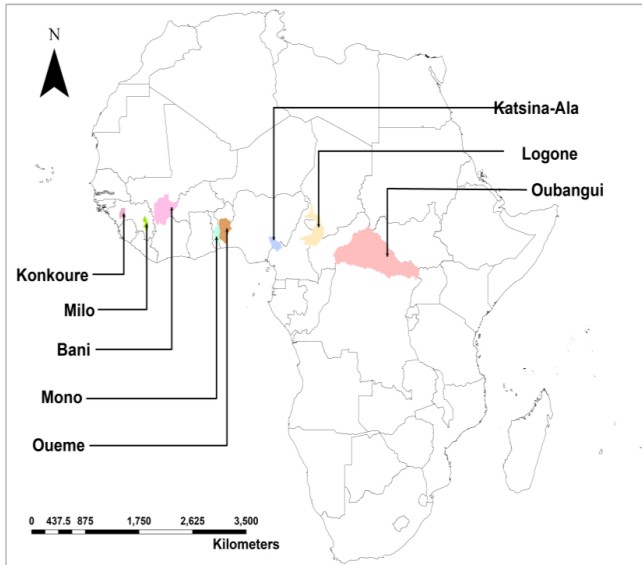

**Figure 1:** Locations of the eight river basins where the performance of WRR and gridded ET
products were evaluated
**Table 1:** Characteristics of river basins and sources of river discharge data

| Basin | Total area (km²) | Transboundary (Yes or No) Countr(y/ies) | Population (thousands) | Source of river discharge data |
|---|---|---|---|---|
| Bani | 101,600 | (**Yes**) Ivory Coast, Mali, and Burkina Faso | 63,766 | GRDC |
| Katsina-Ala | 22,963 | (**Yes**) Cameroon and Nigeria | 219,875 | NHSA |
| Konkoure | 10,250 | (**No**) Guinea-Conakry | 13,053 | GRDC |
| Logone | 87,953 | (**Yes**) Cameroon, Chad, and Central Africa Republic | 44272 | LCBC |
| Milo | 9,620 | (**No**) Guinea-Conakry | 13,053 | GRDC |
| Mono | 21,575 | (**Yes**) Togo, Benin | 21,479 | Co-author |
| Oubangui | 499,000 | (**Yes**) Central Africa Republic and the Democratic Republic of Congo | 88,742 | GRDC |
| Oueme | 46,990 | (**No**) Benin | 11,488 | Co-author |

Global River Discharge Centre [GRDC], Nigeria Hydrological Services Agency [NIHSA], Lake Chad Basin
Commission [LCBC].
**2.2. Input data**

**2.2.1.  Water resources reanalysis [WRR]**

The WRR product evaluated in this study include "The Global Earth Observation for Integrated
Water Resources Assessment" (eartH2Observe), Famine Early Warning Systems Network



[FEWS NET] Land Data Assimilation System (FLDAS), and TerraClimate. The eartH2Observe
Tier 1 product consists of a multi-model ensemble of ten global models at a spatial resolution of
0.5° x 0.5° spanning from 1979 to 2012 and driven by Watch Forcing Data methodology applied
to ERA-Interim reanalysis (WFDEI) data (Schellekens et al., 2017). The WRR from the
eartH2Observe project are freely available through the project data portal
(https://wci.earth2observe.eu/portal/). Model evaluation here omits the Joint UK Land
Environment Simulator (JULES), Simple Water Balance Model (SWBM), and the simple
conceptual HBV hydrological model (HBV-SIMREG) as data from the models was not available
from the data portal for the selected basins at the time of writing. As such, seven models and
model ensemble were included in this study. Although there is an available Tier 2 product with a
higher spatial resolution (0.25°), this study did not utilise these data as selected basins were not
included at the time of conducting this research. We also evaluated the NOAH model from
FLDAS with sptial resolution of 0.1° and runoff data from TerraClimate reanalysis with a spatial
resolution of 0.041°. Table 2 provides a brief summary of the different models used in this study.
**Table 2:** Water resources reanalysis (WRR) evaluated

| Model provider | Model name | Model type | Routing scheme | Reference |
|---|---|---|---|---|
| CNRS (Centre National de la Recherche Scientifique) | ORCHIDEE (Organizing Carbon and Hydrology in Dynamic Ecosystems) | LSM | Cascade of linear reservoirs | (Krinner et al., 2005) |
| CSIRO (Commonwealth Scientific and Industrial Research Organization) | AWRA-L (Australian Water Resources Assessment | GHM | Cascade of linear reservoirs | (Van Dijk et al., 2014) |
| ECMWF (European Centre for Medium-Range Weather Forecasts) | HTESSEL (Hydrology Tiled ECMWF Scheme for Surface Exchanges over Land) | LSM | CaMa-Flood | (Balsamo et al., 2009) |
| JRC (Joint Research Centre) | LISFLOOD | GHM | Double kinematic wave | (Van Der Knijff et al., 2010) |
| UniUt (Universiteit Utrecht) | PCR-GLOBWB | GHM | Travel time | (Van Beek et al., 2011) |
| MeteoFr (Meteo France) | SURFEX | LSM | TRIP with stream | (Decharme et al., 2010) |
| UniK (Universitat Kassel) | WaterGAP | GHM | Manning–Strickler | (Wada et al., 2014) |
| NASA | NOAH | LSM | Soil-layer water and energy balance | (Mcnally et al., 2017) |
| University of California Merced | Water- Balance Model | GHM | Bucket type model | (Abatzoglou et al., 2018) |


### 2.2.2. Evapotranspiration products

The gridded ET products evaluated in this study include FLDAS, GLEAM3.5a & 3.5b,
MODIS16A2, PMLV1, PMLV2, SSEBop, and TerraClimate (see Table 3). Data from the ET



products are freely available with a global coverage except for FLDAS, which covers only the
African domain. Although the gridded ET products all have different spatial resolutions, we did
not resample the data to the same resolution because a previous study has shown that resampling
does not have any significant impact on the results (Weerasinghe et al., 2020). We also leveraged
on the power of cloud computing by downloading data for some ET products using the climate
engine research App. (www.climateengine.com). Table 3 provides a summary of all ET products
evaluated in this study.
**Table 3: Summary of the characteristics of the different ET products**

| ET product | Core equation | Temporal resolution | Spatial resolution | References |
|---|---|---|---|---|
| FLDAS | Penman–Monteith | Daily | 0.1° x 0.1° | (Mcnally et al., 2017) |
| GLEAM3.5a & 3.5b | Priestley-Taylor | Monthly | 0.25° x 0.25° | (Martens et al., 2017) |
| MODIS16A2 | Penman-Montieth | 8-day | 1/48°x1/48° | (Mu et al., 2007; Mu et al., 2011) |
| PMLV1 | Penman–Monteith–Leuning | Monthly | 0.5° x 0.5° | (Zhang et al., 2016) |
| PMLV2 | Penman–Monteith–Leuning | 8-day | 1/192°x1/192° | (Zhang et al., 2019) |
| SSEBop | Surface Energy Balance | Monthly | 1/96° x 1/96° | (Senay et al., 2013) |
| TerraClimate | Penman–Monteith | Monthly | 1/24° x 1/24° | (Abatzoglou et al., 2018) |


### 180    2.3. Evaluation data

#### 181    2.3.1.    River discharge

Observed river discharge data were used to evaluate the performance of WRR models and to
estimate basin-wide water balance evapotranspiration ($ET_{WB}$) using the water balance concept.
The source of the river discharge data is available in Table 1. Gaps in the discharge data were
filled using Self-Organizing Maps which have been shown to be a robust method for infilling
missing gaps in hydrometeorological time series (Nkiaka et al., 2016).

#### 187    2.3.2.    Precipitation

Climate Hazards Group InfraRed Precipitation with Station data (CHIRPS) was used in this study
to estimate $ET_{WB}$. CHIRPS has a quasi-global coverage at a spatial resolution of 0.05° x 0.05°,
spanning the period from 1981 to the present at a daily timescale (Funk et al., 2015). The dataset
was explicitly designed taking into consideration the weaknesses of existing products (Sulugodu
et al., 2019). As such, CHIRPS blends gauge and satellite precipitation covering most global land
regions, it has low latency, high resolution, low bias, and long period of record (Funk et al., 2015).
CHIRPS has extensively been validated (Dinku et al., 2018; Satgé et al., 2020) and used in several
studies in Africa (Larbi et al., 2021; Nkiaka, 2022). The data was downloaded as the spatial
average for each basin using the climate engine App and used to estimate $ET_{WB}$



### 2.3.3. GRACE

GRACE data are monthly anomalies of terrestrial water storage changes (TWSC) used to quantify changes in terrestrial water storage. The dataset has a global coverage spanning the period 2003–2017 (Tapley et al., 2019). The data was derived from Jet Propulsion Laboratory (JPL) RL06M Version 2.0 GRACE mascon solution at a spatial resolution of 0.5° x 0.5°. The data has a coastline resolution improvement (CRI) filter to reduce leakage errors across coastlines and land-grids, using scaling factors derived from the community land model (Wiese et al., 2016). GRACE data has recently been re-processed to reduce measurement errors and represents a new generation of gravity solutions that do not require empirical post-processing to remove correlated errors, as such, the present data is better than the previous GRACE version that was based on spherical harmonic gravity solution (Wiese et al., 2016). GRACE data was used in this study to estimate $ET_{WB}$ following the approach used in several other studies e.g., (Andam-Akorful et al., 2015; Liu, 2018; Xie et al., 2022).

### 2.4. Evaluating gridded WRR

WRR models were evaluated following a multi-objective approach commonly used in evaluating the performance of hydrological models, including the Nash-Sutcliffe efficiency (NSE), Kling-Gupta efficiency (KGE), and the percent bias (PBIAS). NSE scores range from -∞ to 1, with 1 indicating a perfect representation of observed discharge. NSE scores ≥0.50 can be considered acceptable whereas NSE scores ≤0.0 indicate poor model performance (Moriasi et al., 2007). Similar to NSE, the KGE is a dimensionless metric that can be decomposed into three components that are crucial for evaluating hydrological model performance accounting for temporal dynamics (correlation), bias errors (observed vs simulated volumes), and variability errors (relative dispersion between observations and simulations) (Gupta et al., 2009). KGE scores also range from −∞ to 1, with 1 considered the ideal value. Next, PBIAS is used to measure the tendency of the simulated discharge to be larger or smaller than their observed counterparts (Gupta et al., 2009). PBIAS is expected to be 0.0, with low magnitude values indicating accurate simulations, positive values indicate underestimation, negative values indicate overestimation (Moriasi et al., 2007). According to Moriasi et al. (2007), a hydrological model with PBIAS values in the range ±25 % can be considered to be acceptable. Furthermore, a temporal evaluation of flow hydrographs was carried out by plotting the monthly simulated vs observed discharge to ascertain visually if the models were able to capture the magnitude, seasonality, and interannual variability of discharge.





**Table 4: Contingency table for 80th percentile river discharge**

|  |  | Observed discharge | |
|---|---|---|---|
|  |  | Yes | No |
| Simulated discharge | Yes | Hits (H) | False Alarms (FA) |
|  | No | Misses (M) | Correct Negatives |


Lastly, we evaluated the models ability to predict discharge above specific thresholds. This
evaluation step is of critical importance when considering operational water management
requirements such as water allocation and reservoir operation which rely on monthly river
discharge. To achieve this, we adopted the Critical Success Index (CSI) as the metric to evaluate
the ability of each model to simulate discharge exceeding the 20th and 80th percentiles. CSI is
calculated from a two-dimensional contingency table defining the events in which observed and
simulated discharges exceed a given threshold (Thiemig et al., 2015). We used the 20th and 80th
percentiles to assess the ability of the models to simulate both low and high flows respectively.
The contingency table (Table 4) is a performance measure used in summarizing all possible
forecast-observation combinations such as hits (H; event forecasted and observed), misses (M;
event observed but not forecasted), false alarms (FA; event forecasted but not observed) and
correct negatives (CN; event neither forecasted nor observed). The ideal value for CSI is 100%
and the metric is calculated as follows:
$$CSI = \frac{H}{H + M + FA} X\ 100 \qquad (1)$$

**2.5. Evaluating gridded ET**

We also adopted a multi-step approach to evaluate the performance of ET products by assessing
the annual ET–precipitation ratio, evaluating the statistical performance of ET products against
long-term $ET_{WB}$ and the ability of the products to capture monthly ET variability.
In the first step, the annual ET–precipitation ratio was calculated to compare with ratio
obtained from $ET_{WB}$. The ET–precipitation ratio can also provide an estimate of the amount of
water available in each basin after evapotranspiration losses. In the second step, different
statistical metrics were used to assess the performance of the ET products using the monthly
$ET_{WB}$ as a reference (Andam-Akorful et al., 2015; Burnett et al., 2020; Koukoula et al., 2020).
The monthly $ET_{BW}$ was calculated using the basin water balance equation as follows:
$$ET_{WB} = P - Q - \Delta S \qquad (2)$$

Where $P$ is average monthly precipitation over the basin (mm), $Q$ is river discharge (mm) and $\Delta S$
is the terrestrial water storage change [TWSC] (mm). Unlike several studies that have evaluated



ET products on an annual timescale, this study adopts a monthly sample. As such, the TWSC
component ($\Delta S$)  in equation 2 that is often neglected when estimating $ET_{WB}$ over several years
(≥10 years) could not be overlooked. Due to the likely impact of anthropogenic activities such as
reservoir operation, water withdrawal, and monthly rainfall variability on TWSC, values derived
at monthly timescales are important. In this case, TWSC data used in this study were obtained
from GRACE.

Due to the coarse spatial resolution of GRACE, it has been argued that GRACE is not

sensitive at detecting changes in monthly TWSC in small-size basins ≤150,000 km$^2$ (Rodell et
al., 2011). Based on this claim, it might be argued that GRACE data may not be applicable in this
study considering that most of the basins are below this threshold except the Oubangui (499,000
km$^2$). However, several studies (Liu, 2018; Biancamaria et al., 2019; Oussou et al., 2022; Xie et
al., 2022), have demonstrated that GRACE can provide acceptable TWSC estimates for basins
that are smaller than this threshold. Encouraging results from these and other studies do therefore
suggest that GRACE data can be used in this study; albeit with the  expectation of considerable
uncertainties in TWSC estimates. For this study, GRACE data for each basin were obtained by
averaging the timeseries of all coincident GRACE grid cells. To estimate changes in monthly
TWSC, we calculated the difference between consecutive GRACE measurements for each basin,
divided by the time between measurements, using the following equation:
$$\Delta S = (S_{[n]} - S_{[n-1]})/dt \qquad\qquad (3)$$
where $\Delta S$ represents the TWSC (mm), $n$ is the measurement number, and $dt$ is the time difference
between two consecutive GRACE measurements (months).

Lastly, temporal evaluation of the products was carried out by plotting the time series of

all ET products against $ET_{WB}$ to visually establish if the gridded ET products were able to capture
the magnitude, seasonality, and interannual variability of ET across the basins.
**2.6. Estimating relative uncertainty in basin-scale water balance ET ($ET_{WB}$)**
To estimate the relative uncertainty in monthly $ET_{WB}$, we first calculated the absolute uncertainty
in monthly $ET_{WB}$ by propagating errors through each of the components in equation 2 (Rodell et
al., 2011), as follows:
$$\sigma ET = \sqrt{\sigma_{P+}^2 \sigma_{Q+}^2 \sigma_{\Delta S}^2} \qquad\qquad (4)$$
Where $\sigma_P,$ $\sigma_Q$ and $\sigma_{\Delta S}$ represent the absolute uncertainties in basin precipitation, observed river
discharge, and TWSC respectively. Uncertainty in precipitation was estimated as systematic



errors (bias). For this, we used a value of 2 % estimated for CHIRPS data at monthly timescale
from 1981–2016 over Africa from a validation study using the Global Precipitation Climatology
Centre (Shen et al., 2020). Uncertainty in TWSC was determined using the gridded fields of
measurement and leakage errors (residual errors after filtering and rescaling) that are provided
with the GRACE data. The uncertainty for each basin was calculated by averaging the values of
all GRACE grid cells within each basin. To account for month-to-variation in equation 3, the
TWSC error values were multiplied by $\sqrt{2}$ to obtain $\sigma_{\Delta S}$ (Andam-Akorful et al., 2015). Because
no uncertainty estimates were provided with the river discharge data, we adopted a value of 20
% which has been used in a recent study in the region (Burnett et al., 2020). After calculating the
absolute uncertainty in monthly $ET_{WB}$, the relative monthly uncertainty was calculated using
equation 4 (Baker et al., 2021) as follows:
$$\upsilon ET = \frac{\sigma ET}{ET_{WB}} X 100 \qquad\qquad (5)$$
Where $\upsilon ET$ is the monthly relative uncertainty (%), $\sigma ET$ is the absolute monthly uncertainty
(mm), and monthly $ET_{WB}$ (mm). Figure 2 shows a flowchart WRR and ET products evaluation
steps.

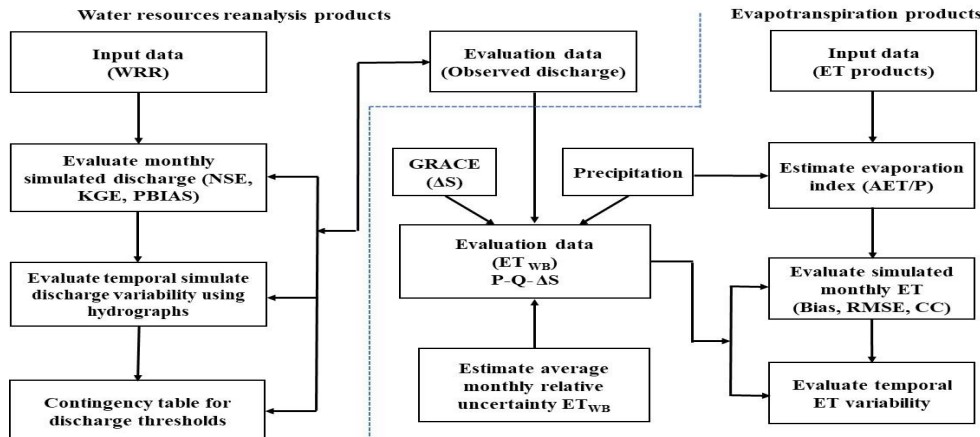


**Figure 2:** Flowchart outlining the steps used in evaluating the WRR and ET products (The blue

dotted line in the flow chart separates evaluation of WRR from ET products)

**3.   Results**
**3.1. Water resources reanalysis products**

**3.1.1.   Hydrological performance**

A multi-objective approach using different statistical metrics (NSE, KGE and PBIAS) was used
to evaluate the models in WRR Tier 1. The performance of the models in simulating river





discharge is shown in Figure 3. Using the NSE as a performance metric, results show that NOAH
produced positive scores in all the basins (0.15–0.48). Terra, AWRAL and Lisflood models
produced positive scores (0.01–0.75) in seven, six and four basins respectively. SURFEX model
produced positive scores in three basins while ORCHIDEE, HTESSEL, Watergap and the
ensemble mean produced positive scores in two basins each. PCR-GLOBW produced negative
scores in all the basins (Figure 3a).

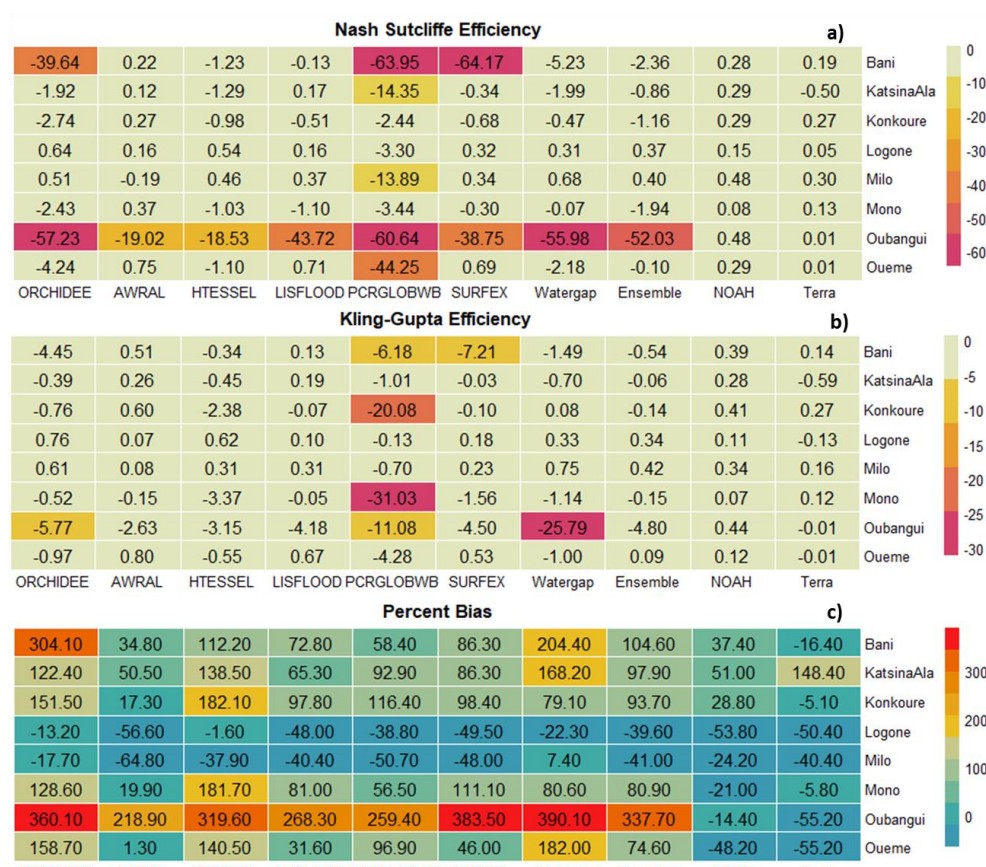


**Figure 3:** Statistical evaluation of the models using (a) NSE, (b) KGE, and (c) PBIAS. Red and
orange colours represent poor model performance in Figures 3a, 3b & 3c, however, the acceptable
PBIAS range in Figure 3c is ±2 5%. Ensemble refers to the mean of WRR from the
earthH2Observe product.
Results of the KGE show that NOAH also produced positive scores (0.11– 0.44) in all basins,
followed by AWRAL, Lisflood and Terra models with positive scores in six, five and four basins
respectively (Figure 3b). SURFEX and Watergap produced positive scores in three basins
ORCHIDEE while HTESSEL produced positive scores (0.31–0.76) in two basins, the ensemble





mean produced positive scores (0.09 – 0.42) in three basins. PCRGLOBW produced the worse
KGE scores (Figure 3b).

Positive and negative PBIAS values were obtained in the different basins. Negative values

indicate that the model overestimated discharge volumes compared to observed discharge while
positive values indicate the opposite. NOAH, Terra and AWRAL produced acceptable PBIAS
scores (±25 %) in three basins, ORCHIDEE and Watergap produced similar scores in two basins
and HTESSEL in one basin (Figure 3c). The rest of the models including the ensemble mean
either grossly overestimated or underestimated discharge volumes in all the basins.

### 3.1.2. Temporal evaluation

The ability of the models to capture discharge variability was analysed by comparing the
simulated vs observed discharge in all the basins. Results show that most models were able to
capture the seasonal discharge variability including peak and low flows (Figure 4). However,
PCR-GLOBW systematically overestimated low flows and underestimated high flows across all
basins. In the Oubangui basin, all models were able to capture the seasonal variability but
consistently underestimated peak flows except NOAH and Terra models which both
overestimated peak flows (Figure 4). For example, peak discharge in the river exceeds 5000
m$^3$/sec, but all models except NOAH and Terra simulated this peak discharge to be less than 2000
m$^3$/sec (Figure 4).

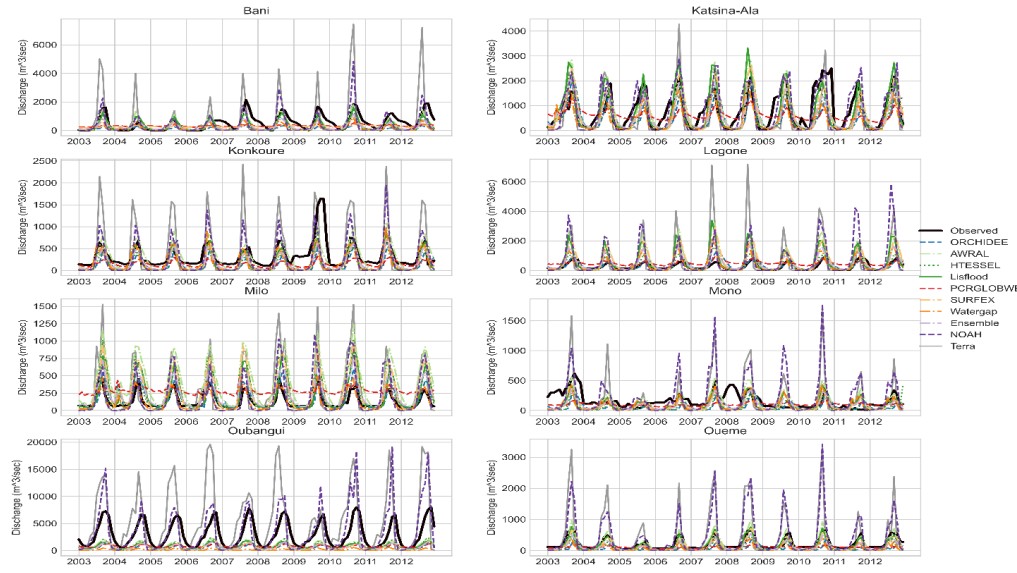


**Figure 4:** Evaluation of temporal flow variability simulated by the different model





### 3.1.3. Critical Success Index

Figure 5 shows the performance of the models in simulating the $80^{th}$ and $20^{th}$ percentiles

monthly discharge. For the $80^{th}$ percentile flows, results show that NOAH and Terra produced
CSI scores above 50 % in all basins followed by Lisflood and AWRAL in seven and six basins
respectively while Surfex and Watergap produced similar scores in four basins each (Figure
5a). For the $20^{th}$ percentile flows, only NOAH produced CSI scores above 50 % in four basins
while Lisflood produced similar scores in two basins. The performance of the other models in
simulating the $80^{th}$ percentile flow shows a large spread while most models including the
ensemble mean failed to simulate the $20^{th}$ percentile flow across all the basins. Taking together,
results suggest that the models simulated high flows better than the low flows with only the
NOAH model capable of capturing both flow regimes in most basins (Figure 5b).

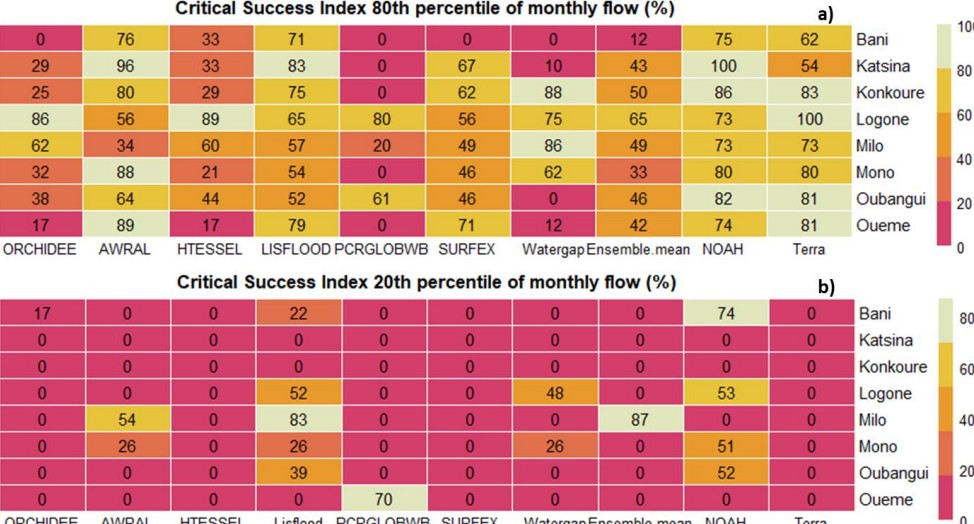

**Figure 5:** Critical Success Index for $80^{th}$ and $20^{th}$ percentile of monthly flow across all basins

### 3.2. Evapotranspiration products

#### 3.2.1. Evapotranspiration–precipitation ratio

Figure 6 shows the annual ET–precipitation ratio for all basins. It can be observed that average
annual ET–precipitation ratio ranges between 0.52–0.82 over a period of 10 years (2003–2012)
across all basins. SSEBop produced the highest ET–precipitation ratios (0.53–0.99) while
MOD16A2 produced the lowest ratio (0.41–0.66) (Figure 5). Results show that the evaporation
ratios from most of the ET products are in the same order of magnitude with the ratio from





$ET_{WB}$ across all the basins with the only exception being SSEBop and MOD16A2 which
respectively overestimated and underestimated this value.

**Evaporation - Precipitation ratio**

| ETWB | FLDAS | Gleam35a | Gleam35b | MOD16A2 | PMLV1 | PMLV2 | SSEBop | Terra | |
|------|-------|----------|----------|---------|-------|-------|--------|-------|---|
| 0.63 | 0.74 | 0.66 | 0.69 | 0.45 | 0.77 | 0.77 | 0.99 | 0.85 | Bani |
| 0.44 | 0.48 | 0.47 | 0.42 | 0.41 | 0.51 | 0.51 | 0.53 | 0.48 | KatsinaAla |
| 0.46 | 0.54 | 0.53 | 0.57 | 0.48 | 0.71 | 0.62 | 0.76 | 0.57 | Konkoure |
| 0.69 | 0.71 | 0.59 | 0.60 | 0.44 | 0.67 | 0.72 | 0.89 | 0.74 | Logone |
| 0.54 | 0.65 | 0.54 | 0.56 | 0.51 | 0.63 | 0.59 | 0.73 | 0.61 | Milo |
| 0.65 | 0.68 | 0.69 | 0.71 | 0.63 | 0.67 | 0.73 | 0.86 | 0.78 | Mono |
| 0.72 | 0.70 | 0.68 | 0.68 | 0.66 | 0.66 | 0.77 | 0.86 | 0.67 | Oubangui |
| 0.65 | 0.69 | 0.66 | 0.68 | 0.60 | 0.69 | 0.74 | 0.98 | 0.71 | Oueme |
| 0.60 | 0.65 | 0.60 | 0.61 | 0.52 | 0.66 | 0.68 | 0.82 | 0.68 | Average |

**Figure 6:** Annual evapotranspiration – precipitation ratio 2003 – 2012
### 3.2.2. Basin-wide water balance estimates
Figure 7 shows the results of the statistical metrics used in evaluating the ET products using
monthly $ET_{WB}$ as a reference. Considering bias as a performance metric, several products e.g.,
FLDAS, PMLV2, Terra, and GLEAM3.5a &3.5b produced low bias scores ranging from -6 to
11 mm/month. However, GLEAM products systematically underestimated monthly ET with
respect to $ET_{WB}$ in all the basins while FLDAS, Terra and PMLV2 produced mixed results
(7a). While SSEBop systematically overestimated monthly ET in all the basins, MODIS16A2
underestimated this variable in all but one basin with respect to monthly $ET_{WB}$ (Figure 7a). The
lowest bias values ranging from -8.30 to 13.37 mm/month were obtained in the Katsina-Ala
basin while the highest bias values ranging from -14.61 to 26.33 mm/month were recorded in
the Konkoure basin.

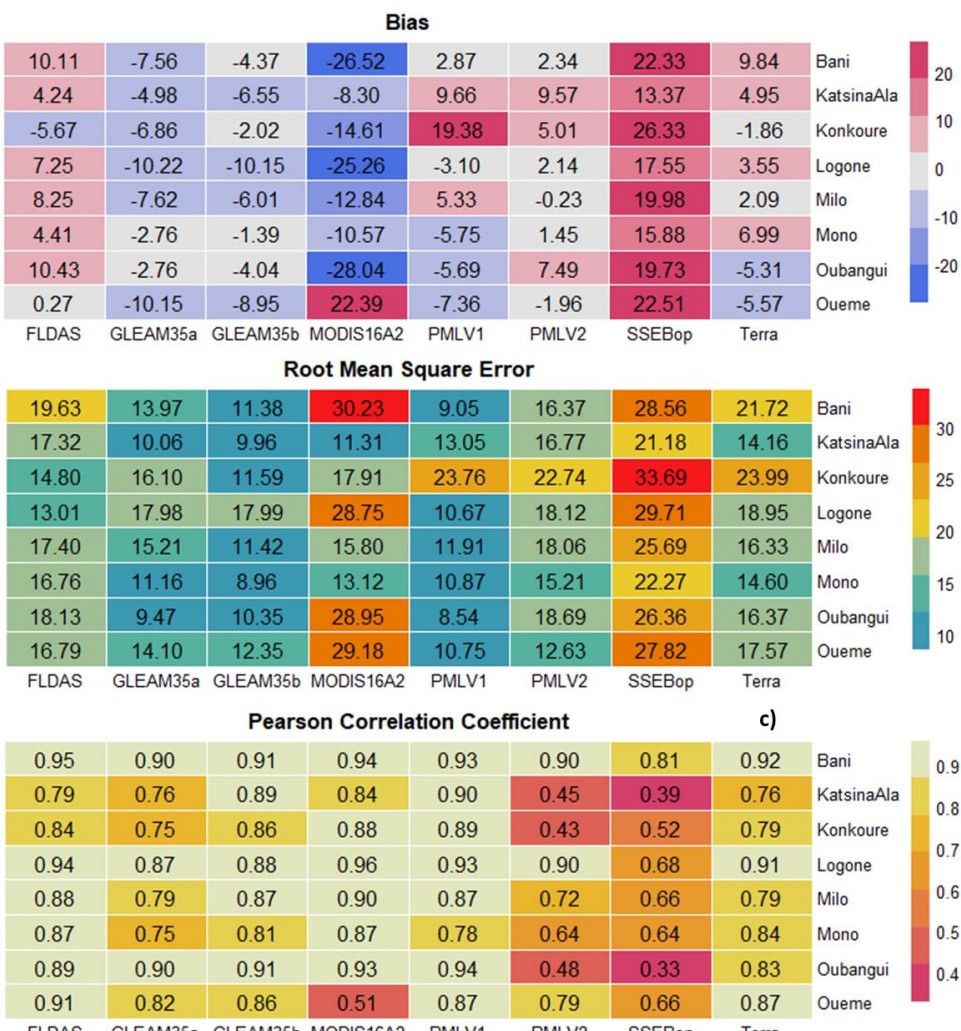

**Figure 7:** Bias, RMSE, and Pearson correlation coefficient between monthly $ET_{WB}$ and different ET products.

GLEAM3.5a & b produced the lowest RMSE (9.47–18 mm/month), followed by FLDAS (13–20 mm/month) and PMLV1 (8.50–12 mm/month) with this score exceeding 20 mm/month in only one basin. The rest of the ET products produced substantially higher RMSE scores with SSEBop and MODIS16A2 producing the highest RMSE scores (Figure 7b). Most ET products produced high Pearson correlation scores (≥0.75) with respect to $ET_{WB}$ in all basins except PMLV2 and SSEBop which both produced low scores (<0.50) in three and two basins respectively (Figure 7c).

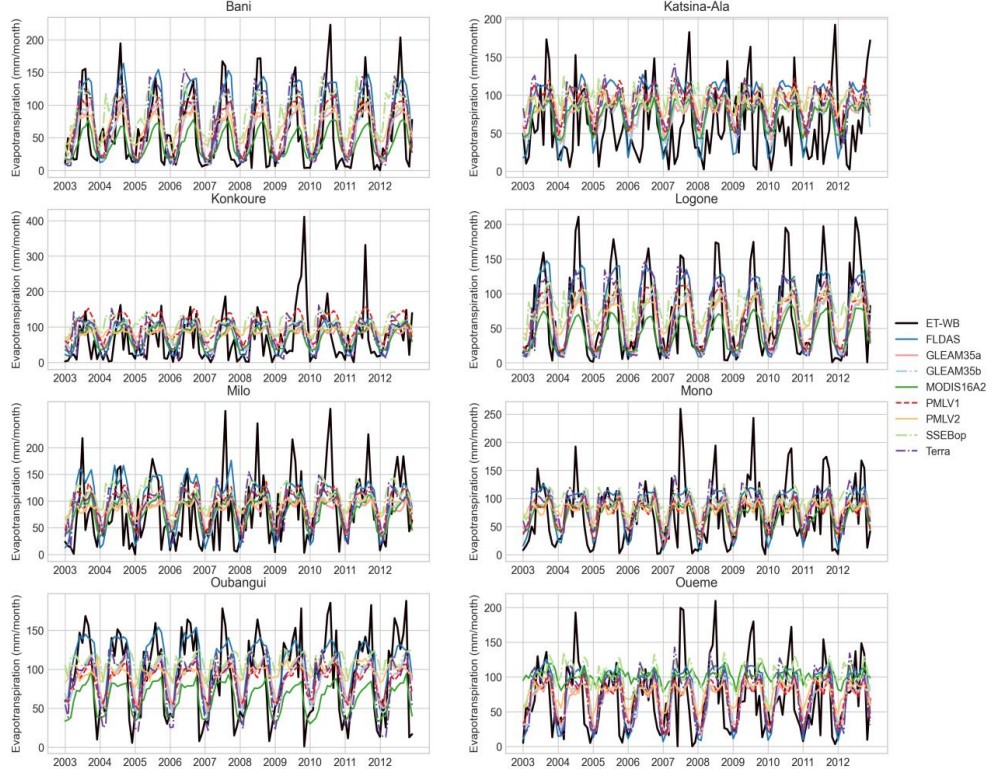

**Figure 8:** Seasonal cycle of ET products and basin-wide water balance evapotranspiration. $ET_{WB}$ represents monthly evapotranspiration estimated by the water balance method, while the rest are model-derived ET products.

### 3.2.3. Monthly ET variability

Figure 8 shows the seasonal cycle of $ET_{WB}$ against the ET products for all basins. It can be observed that most products were able to replicate the seasonal ET cycle across the basins. However, most ET products underestimated monthly ET compared to $ET_{WB}$ during the rainy season with MOD16A2 producing the poorest results. Furthermore, most products were not able to replicate the high peaks produced by $ET_{WB}$ during the rainy season,.

### 3.2.4. Estimating relative uncertainty in $ET_{WB}$

An assessment of absolute uncertainties in monthly $ET_{WB}$ indicated that the dominant sources of uncertainty vary from one basin to another and by each month. For example, in the Katsina-Ala, Konkoure, and Milo basins, the dominant source of uncertainty in monthly $ET_{WB}$ was from river discharge (**Appendix A**). Although the absolute uncertainty in precipitation and TWS also appears to be substantial in the three river basins, the uncertainty in river discharge





takes precedence over the other sources of uncertainty due to its higher magnitude. On the
contrary, the dominant source of uncertainty in $ET_{WB}$ in the Bani, Logone, and Oubangui basins
was from TWSC. It can also be observed across the basins that there was no significant
variation in monthly TWSC uncertainty which is consistent with the results of a similar study
in the Amazon basin (Baker et al., 2021). Results also revealed that the magnitude of TWSC
uncertainty were similar across the basins irrespective of the basin size (**Supplementary**
**material**).

Figure 9 shows the relative uncertainty in $ET_{WB}$ across all the basins. It can be observed

that relative uncertainty values are generally <30 % but vary from month to month. However,
the values were exceptionally high in the Katsina-Ala and Konkoure basins. The relative
uncertainty in $ET_{WB}$ also appears to be exceptionally high in the months of September–
November which corresponds to high flow season. Taking together, the average monthly
relative uncertainty in $ET_{WB}$ for all basins ranges from 10–18% except in the Katsina-Ala and
Konkoure basins where this range is grossly exceeded.

### Relative Uncetainty ET-WB

| Bani | KatsinaAla | Konkoure | Logone | Milo | Mono | Oubangui | Oueme | |
|---|---|---|---|---|---|---|---|---|
| 28.63 | 46.60 | 63.25 | 23.65 | 24.55 | 22.40 | 11.71 | 18.12 | January |
| 14.59 | 49.90 | 39.58 | 19.50 | 11.22 | 13.04 | 11.18 | 11.77 | February |
| 19.70 | 68.50 | 30.39 | 17.88 | 14.07 | 9.07 | 7.91 | 9.16 | March |
| 10.48 | 39.91 | 33.78 | 11.29 | 10.91 | 6.75 | 4.98 | 5.96 | April |
| 8.51 | 79.73 | 18.60 | 4.77 | 7.92 | 6.59 | 4.84 | 4.92 | May |
| 5.81 | 53.77 | 13.25 | 6.77 | 6.99 | 5.06 | 4.74 | 5.40 | June |
| 3.99 | 40.73 | 21.01 | 3.50 | 5.83 | 4.22 | 4.14 | 4.21 | July |
| 4.35 | 118.49 | 21.41 | 3.70 | 14.46 | 5.00 | 4.54 | 5.71 | August |
| 11.58 | 57.43 | 59.88 | 6.31 | 18.65 | 6.48 | 6.52 | 6.82 | September |
| 34.40 | 28.00 | 26.00 | 15.01 | 14.86 | 9.35 | 5.86 | 11.48 | October |
| 51.86 | 31.79 | 27.69 | 57.36 | 30.83 | 43.82 | 25.06 | 25.44 | November |
| 20.21 | 28.20 | 33.64 | 17.22 | 15.31 | 28.22 | 24.62 | 15.41 | December |
| 17.84 | 53.59 | 32.37 | 15.58 | 14.63 | 13.33 | 9.67 | 10.37 | Mean |

**Figure 9:** Average (2003 – 2012) monthly relative uncertainty in monthly $ET_{WB}$ (%)

### 4. Discussion

The overarching goal of this paper was to assess the performance of gridded water resources
reanalysis and evapotranspiration products and to estimate the relative uncertainty in monthly
basin-wide evapotranspiration ($ET_{WB}$) estimates. Below we provide a discussion and
implications of our results in water security assessment in ungauged basins.

### 4.1. Water resources reanalysis

The performance of WRR products was assessed through commonly used model evaluation metrics, discharge variability, and verification skill scores (critical success index) using observed river discharge data. Our results show strong differences in the performance of the different models in simuating river discharge across the basins. NOAH model produced positive NSE and KGE values in all basins and PBIAS values within the acceptable range (±25%) in three basins. Temporal evaluation of the WRR products showed that NOAH, Terra, AWRAL and Lisflood were able to capture the seasonal variability in discharge as demonstrated by high KGE scores. Indeed, high KGE values suggest that some models were able to capture the temporal dynamics (strong correlation), and low bias scores indicate that the variability errors between the observed discharge and simulation was also low (Gupta et al., 2009). Nevertheless, Terra consistently overestimated peak flows in all the basins.

Apart from NOAH model which is a LSM used in FLDAS, most GHMs used in earthH2Observe tier 1 product performed better than the LSMs, which is consistent with results from other studies (Lakew et al., 2020). The strong performance of GHMs compared to LSMs can be attributed to the differences in the model structure and parametrisation schemes between LSMs and GHMs (Gründemann et al., 2018; Koukoula et al., 2020). For example, some GHMs such as Watergap are able to simulate lakes and reservoirs and water withdrawal while LSMs can only simulate natural processes. Such differences in model structure can significantly influence discharge volumes simulated by both types of models (Gründemann et al., 2018). Although PCRGLOBW is a GHM, it produced substantially low performance compared to the LSMs which is consistent with results from other studies in the region (Gründemann et al., 2018; Lakew et al., 2020). This suggest that PCRGLOBW model may not be suitable for assessing water security in the region.

The ability of the models to simulate flow thresholds was evaluated using the CSI. Results show that NOAH, Terra, AWRAL and Lisflood were able to capture more than 50% of 80[th] percentile monthly flow in most basins. We also noted that apart from NOAH model, the rest of the GHMs performed better than the LSMs from eartH2Observe in their ability to capture the 80[th] percentile monthly flows across the basins while only NOAH was able to capture 20[th] percentile flows in three basins. The better performance of NOAH model compared to other models evaluated in this study can be attributed to the fact that FLDAS was specially designed and optimized to produce physically meaningful quantitative data for monitoring food and water security in data-scarce regions in Africa (Mcnally et al., 2017). The slight better performance of NOAH can also be attributed to its higher spatial resolution (0.1°)

compared to other models with coarser spatial resolution (0.5°). Terra with a spatial resolution
of 0.041° also performed slightly better than the other models with coarser spatial resolution.
In fact, a previous study (Gründemann et al., 2018), has shown that WRR products with higher
spatial resolution perform better than products with coarser resolution in their ability to
simulate discharge. The better performance of NOAH can also be attributed to the fact the
FLDAS is driven by a combination of different precipitation products which reduces the
uncertainties in the input data while earth2oberve tier 1 product are driven by only one data
source (WFDEI) with uncertainties in the input data which is propagated to the model outputs.
Our results also showed that Lisflood performed better than most other earth2oberve models
which can also be attributed to the fact that Lisflood has been extensively used in research and
operational settings in Africa (Thiemig et al., 2015; Smith et al., 2020). As such, the  model
parameters may have been better constrained in the region than other models from
eartH2Observe. Taking together, results from this study highlight the importance of evaluating
outputs from WRR products in representative basins before applying them in studies that may
have wider policy and financial implications. However, our results suggest a need to enhance
the spatial resolution of WRR products and for these products to be driven by data from
multiple sources to reduce the uncertainties input data.
**4.2. Evapotranspiration products**
The annual ET – precipitation ratio produced by the ET products in this study is in the same
order of magnitude with that produced by $ET_{WB}$ except for SSEBop and MOD16A2 which are
within the range estimated for the global land regions (Rodell et al., 2015). This indicates that
most ET products performed well in this aspect of the ET evaluation. The annual ET –
precipitation ratios obtained in this study suggests that annual ET does not exceed annual
precipitation in any of the basins during the period under evaluation which is an indication of
available water resources in each basin.

Taking together all the ET evaluation criteria, FLDAS, GLEAM3.5a & 3.5b, Terra and

PMLV2 appear to outperform the other products even though GLEAM products systematically
underestimated ET in all the basins. Conversely, SSEBop and MOD16A2 produced poor did
not perform well in all the basins and may not be suitable for water security assessments in the
region. Our results are generally consistent with those from other studies indicating that
GLEAM and MODIS16A2 underestimate evapotranspiration, while SSEBop overestimates
this variable in most parts of Africa (Weerasinghe et al., 2020; Adeyeri and Ishola, 2021;
Mcnamara et al., 2021). Given that FLDAS ET estimate is derived from a LSM (NOAH) with



other water balance components (runoff, soil moisture and baseflow), it may be more preferable
for assessing water security in ungagued basins because of water balance closure. Our results
also revealed that the performance of the ET products was not influenced by spatial resolution
which is consistent with results from previous studies (Weerasinghe et al., 2020; Jiang and Liu,
2021). For example, Gleam products with a spatial resolution of 0.25° outperformed products
such as MODIS16A2 and SSEBop with higher spatial resolutions. Weerasinghe et al. (2020)
reported that re-gridding ET products to the same spatial resolution did not have any significant
impact on the performance of the product.
Although all the products were able to capture the temporal ET cycle in the basins, there
were substantial differences in the magnitude of monthly ET from each model. This finding is
consistent with results from other studies showing strong differences in ET estimates produced
by different models over Africa (Weerasinghe et al., 2020; Adeyeri and Ishola, 2021). The
discrepancies in monthly ET estimates from the models may be attributed to differences in the
equations underpinning each ET model, model parameters, and uncertainties in the input data
used in driving the models. This is also in-line with findings from another study in West Africa
highlighting the impact of model parameters and precipitation input uncertainty on ET
estimates (Jung et al., 2019). Considering the aforementioned factors, it may be difficult to
expect the products to produce similar results. $ET_{WB}$ estimates across all the basins produced
very high peaks during the rainy season which is also similar to the results of a related study in
West Africa (Andam-Akorful et al., 2015). The high peaks observed in $ET_{WB}$ may be attributed
to errors inherent in monthly precipitation, river discharge, and TWSC estimates used in
estimating monthly $ET_{WB}$.
Given that there was no uncertainty information on the river discharge data used in this
study, we adopted a value of 20 % following a previous study in the region (Burnett et al.,
2020). In fact, we feel that this value is conservative considering that uncertainties in river
discharge in tropical regions have been shown to range from 41 to 200 % (Kiang et al., 2018).
The mean monthly relative uncertainty for $ET_{WB}$ for most basins appears to be in the same
order of magnitude (16 %) with results obtained in the Amazon basin (Baker et al., 2021).
Results also showed that  the relative uncertainty in $ET_{WB}$ is not influenced by basin size as
both large and small basins produced similar (same order of magnitude) uncertainty estimates.
Relative uncertainty in monthly $ET_{WB}$ was higher during the rainy season. This can be linked
to high rainfall input during the rainy season which translates to high river discharge and TWSC
thereby increasing the absolute uncertainties in the different water balance components terms
used in estimating monthly $ET_{WB}$. Another study has shown that rainfall input is a major source
of uncertainty in river discharge due to its sensitivity to rainfall changes (Berghuijs et al., 2017).
Results from this study suggest that the relative the uncertainty in monthly $ET_{WB}$ may be
substantial which may influence the performance scores of the ET products when they are
evaluated using the $ET_{WB}$ method. We therefore recommend that evaluating the performance
of ET products at this monthly timescale should be accompanied with the estimataion of
relative uncertainties in monthly $ET_{WB}$.
**5.  Conclusions**
The objectives of this study were to assess the performance of water resources reanalysis and
evapotranspiration products and to estimate the relative uncertainties in monthly $ET_{WB}$ across
eight basins in Africa**.** Results show varying strengths and weaknesses for the different models
used in the WRR products. Some models were able to capture the river discharge dynamics in
the basins while other models could not adequately capture this patter. Differences in the model
performance can be attributed to differences model structure, parameters, input data used in
driving the models and the spatial resolution of the WRR products. Apart from NOAH which
is a land surface model (LSM), global hydrological models (GHMs) evaluated in this study
performed better than LSMs while PCRGLOBW which is a GHM did not perform well.
Evaluation of gridded ET products also revealed varying strengths and weaknesses for
the different products. Based on the different evaluation criteria (bias, RMSE, Pearson
correlation coefficient, and temporal ET variability), FLDAS appears to outperform most of
other ET products and may therefore be recommended for water security assessment in the
region. More so, because of water balance closure and the availability of other water balance
components (runoff, soil moisture and baseflow). Our results also suggest that the performance
of the ET products is not influenced by spatial resolution, while differences in monthly ET
estimates may be attributed to differences in the equations underpinning each ET model and
the sources of input data used to drive the model. We also observed that while spatial resolution
may have an impact on the performance of WRR products, this was not the case with ET
products as their performance appears to not be dependent on the spatial resolution.
Our results also revealed that relative uncertainties in monthly $ET_{WB}$ were substantially
higher during the rainy season which can be attributed to uncertainties emanating from higher
rainfall input leading to an increase in discharge magnitude and TWSC during this period.
Results also revealed that uncertainty in river discharge is the dominant source of uncertainty
in $ET_{WB}$. This underscores the need to prioritize the installation of new gauging stations while
upgrading existing stations because such large uncertainties could constrain our ability to

understand hydrologic variability and flow forecast and could seriously undermine the evaluation results of WRR and ET products and the calibration of hydrological models.

Results from this study suggest that WRR and ET products may be used for water security assessment in ungauged basins. However, it is imperative to evaluate the performance of these products in representative gauged basins before applying them in ungauged basins. This is because applying the products in ungauged basins without evaluating their performance may lead to poor water management decisions with wider policy and financial implications. However, there is also a need for WRR and ET products to be driven by input data from multiple sources to reduce uncertainties in the input data while the spatial resolution of WRR products also need to be enhanced. Results from this study may be used by the products developers to improve on the quality of future generations of WRR and ET products.

**Author contributions:** EN and RGB designed the methodological framework and contributed to the entire strategic and conceptual framework of the study. EN prepared the data, performed the analyses, interpreted the results and wrote the original draft. JN and EIB provided discharge data for the Mono and Oueme basins respectively. All authors read the paper and provided feedback.

**Competing interests**: The authors declare that they have no conflict of interest.

**Acknowledgements:** E.N. was funded by the Leverhulme Trust Early Career Fellowship – Award Number ECF–097–2020. We are grateful to Coralie Adams at Manchester University for writing the Python code that was used to produce Figures 4 & 8.

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
