# Peer review of "Evaluating the accuracy of gridded water resources reanalysis and evapotranspiration products for assessing water security in poorly gauged basins 2"

_Hydrology and Earth System Sciences, 2022_

## Author Comment (AC2)

[revised manuscript text omitted]
        | ORCHIDEE (Organizing    | LSM        | Cascade of linear  | (Krinner et al.,    |
| National de la      | Carbon and Hydrology in |            | reservoirs         | 2005)               |
| Recherche           | Dynamic Ecosystems)     |            |                    |                     |
| Scientifique)       |                         |            |                    |                     |
| CSIRO               | AWRA-L (Australian      | GHM        | Cascade of linear  | (Van Dijk et al.,   |
| (Commonwealth       | Water Resources         |            | reservoirs         | 2014)               |
| Scientific and      | Assessment              |            |                    |                     |
| Industrial Research |                         |            |                    |                     |
| Organization)       |                         |            |                    |                     |
| ECMWF (European     | HTESSEL (Hydrology      | LSM        | CaMa-Flood         | (Balsamo et al.,    |
| Centre              | Tiled ECMWF Scheme      |            |                    | 2009)               |
| for Medium-Range    | for Surface             |            |                    |                     |
| Weather Forecasts)  | Exchanges over Land)    |            |                    |                     |
| JRC (Joint Research | LISFLOOD                | GHM        | Double kinematic   | (Van Der Knijff et  |
| Centre)             |                         |            | wave               | al., 2010)          |
| UniUt (Universiteit | PCR-GLOBWB              | GHM        | Travel time        | (Van Beek et al.,   |
| Utrecht)            |                         |            |                    | 2011)               |
| MeteoFr (Meteo      | SURFEX                  | LSM        | TRIP with stream   | (Decharme et al.,   |
| France)             |                         |            |                    | 2010)               |
| UniK (Universitat   | WaterGAP                | GHM        | Manning–Strickler  | (Wada et al., 2014) |
| Kassel)             |                         |            |                    |                     |
| NASA                | Noah                    | LSM        | Soil-layer water   | (Mcnally et al.,    |
|                     |                         |            | and energy balance | 2017)               |

[revised manuscript text omitted]

---

## Author Comment (AC5)

Dear Editor-in-Chief,

We wish to thank you for offering us another chance to revise our manuscript (**hess-2022-185**). We detail below all of the revisions that we have undertaken in response to the comments of each reviewer.

In line with changes that we have now made to the manuscript as a whole, we also wish to propose a revision to the title to instead read as: "Evaluating the accuracy of gridded water resources reanalysis and evapotranspiration products for assessing water security in ungauged basins". We hope that these revisions are now acceptable.

With kind regards

Elias Nkiaka (on behalf of the co-authors).

**Response to reviewer 1 comments**

1. Design of the study: The authors need to provide a better explanation of why they decided to evaluate runoff and evaporation from completely different sets of models. All GHMs and LSMs provide estimates of all water balance components, especially when the authors consider GHMs and LSMs as a reanalysis product. Currently, these two parts of the paper are totally distinct from each with no connection to each other. If the objective is to assess water security, I would imagine the end-user would be interested in using estimates of all water balance components from one model or a specific ensemble of models.

    **Response:** Thanks for highlighting this flaw in our study. We have now included the evaluation of evapotranspiration estimates from GHMs and LSMs separately from the results of remote sensing-based evapotranspiration estimates. The results are presented as distinct figures and discussed separately in the revised manuscript. Furthermore, we have highlighted in the manuscript the fact the users' needs for the application of ET estimates may vary. L121-124, revised manuscript.

2. Related to the above comment, if water security is the main intention, would not subsurface water availability be an important variable as well? The authors need to justify only evaluating evaporation and runoff. I am sure most GHMs and LSMs provide data of water storage change.

    **Response:** Thanks for this remark. Yes, most GHMs and LSMs provide data for subsurface water. However, we did not evaluate subsurface water availability in this study because of a lack of in situ data that can be used to validate model simulations. We are aware that other studies such as (Koukoula, M., Nikolopoulos, E. I., Dokou, Z., and Anagnostou, E. N.: Evaluation of global water resources reanalysis products in the upper Blue Nile River Basin, Journal of Hydrometeorology, 21, 935-952) used data assimilation methods to estimate changes in terrestrial water storage from WRR. In this study, we limited our evaluation to

discharge and evapotranspiration estimates. In addition, water storage change was used as a variable in estimating basin-scale evapotranspiration.

3. The authors claim that the utility of gridded datasets have not been sufficiently explored in Africa. I do not agree with the claim - authors have ignored the innumerable studies which have used gridded datasets for model calibration, forcings and validation. In fact, gridded evaporation products are routinely used for improving large scale models for African watersheds (Dile et al. 2020, Dembele et al. 2020). The authors themselves have cited many studies which evaluate these datasets over African basins.

**Response:** Thanks for this remark. I beg to differ with this claim. I said and I quote "Whilst the use of outputs from WRR in water management has gained significant attention in many ungauged areas such as Asia and Latin America, they remain largely under-utilized in Africa. For example, there are only a few case studies reporting on the use of these products in the Upper Blue Nile River basin and the Zambezi River basin. On the other hand, several studies evaluating the performance of gridded hydrometeorological variables in Africa have focused mostly on precipitation while a few studies that have evaluated gridded ET products focused on large basins". See L97 – 111, revised manuscript.
Nevertheless, we believe that evaluating the different datasets across several basins of varying sizes will contribute to the contemporary debate on the performances of the different products across Africa.

4. Methodology: The authors do not make a convincing case for comparing the evaporation datasets with water balance-based evaporation estimates, especially (according to the results) when the uncertainties are large. In fact, achieving water balance closure with different sources of P, ET, and TWS is not a trivial task (Lorenz et al. 2015, Koppa et al. 2021, Pan et al. 2012) and is definitely not robust if only one source of data is used for each component.

**Response:** Thanks for highlighting this issue. The use of water balance-based evapotranspiration estimates for validating global evapotranspiration estimates is a well-established technique in hydrology including in gauged and ungauged basins. Few examples of such studies that have applied the water balance concept to evaluate evapotranspiration estimates at basin-scale include:

1. Weerasinghe, I., Bastiaanssen, W., Mul, M., Jia, L., and Van Griensven, A.: Can we trust remote sensing evapotranspiration products over Africa? Hydrology and Earth System Sciences, 24, 1565-1586.
2. Baker, J. C., Garcia-Carreras, L., Gloor, M., Marsham, J. H., Buermann, W., da Rocha, H. R., Nobre, A. D., de Araujo, A. C., and Spracklen, D. V.: Evapotranspiration in the Amazon: spatial patterns, seasonality, and recent trends in observations, reanalysis, and climate models, Hydrology and Earth System Sciences, 25, 2279-2300
3. Blatchford, M. L., Mannaerts, C. M., Njuki, S. M., Nouri, H., Zeng, Y., Pelgrum, H., Wonink, S., and Karimi, P.: Evaluation of WaPOR V2 evapotranspiration products across Africa, Hydrological processes, 34, 3200-3221
4. Liu, W.: Evaluating remotely sensed monthly evapotranspiration against water balance estimates at basin scale in the Tibetan Plateau, Hydrology Research, 49, 1977-1990

We acknowledge that the uncertainties are large and that is why we decided to identify the dominant sources of uncertainties in this study which is not the case in several studies that have used basin-scale water balance estimates to validate global evapotranspiration datasets. We

believe that identifying the sources of uncertainties is a first step towards reducing them and also to inform policy decisions.

We applied only CHIRPS precipitation estimates in this study because a recent study has provided an average of the uncertainty estimates inherent in monthly CHIRPS estimates across the world including the African continent. However, we are aware that there may be regional differences in the uncertainty estimates across Africa. Nevertheless, we believe that our approach is robust and there is no method that is free from uncertainties.

1. Shen, Z., Yong, B., Gourley, J. J., Qi, W., Lu, D., Liu, J., Ren, L., Hong, Y., and Zhang, J.: Recent global performance of the Climate Hazards group Infrared Precipitation (CHIRP) with Stations (CHIRPS), Journal of Hydrology, 591, 125284

We are also aware that GRACE data is processed and made available by three different research centres. We decided to use estimates from Jet Propulsion Laboratory as it is one of the most commonly used GRACE datasets. Moreover, GRACE estimates from Jet Propulsion Laboratory are provided with uncertainties estimates which for each grid point. For each basin, we averaged the uncertainty estimates for all grid points located within the basin to estimate the GRACE uncertainty for that basin. We also wish to highlight to the author that every study adopts different methods, and we believe we have provided sufficient justifications and clarifications on our approach and methods.

The author may also wish to refer to the following article on GRACE estimates produced by JPL.

Wiese, D. N., Landerer, F. W., and Watkins, M. M. (2016). Quantifying and reducing leakage errors in the JPL RL05M GRACE mascon solution, Water Resources Research, 52, 7490-7502, https://doi.org/10.1002/2016WR019344.

5. Despite previous studies using GRACE at higher resolution, I have serious doubts about the applicability of TWS estimates for basins as small as 9,620 sq.km (an order of magnitude smaller than intended GRACE footprint).

**Response:** Thanks for highlighting this issue. I believe we raised this issue in the manuscript and provided a few examples where GRACE data was used in catchments smaller than the size highlighted by the reviewer e.g.,

1. Liu, W.: Evaluating remotely sensed monthly evapotranspiration against water balance estimates at basin scale in the Tibetan Plateau, Hydrology Research, 49, 1977-1990.

Moreover, in each of our basins, there was at least one GRACE grid point located within each basin which was used to represent the TWSC for the whole basin. Where there were two or more GRACE grid points, we calculated the average of all the grid points located within the basin.

6. In summary, the above two points casts serious doubts on the robustness of the ETwb estimates and its use as a reference dataset for evaluating other datasets.

**Response:** We believe that we have provided sufficient justifications on the use of $ET_{WB}$ as a reference data for evaluating ET estimates derived from different sources. We wish to

reiterate to the reviewer that this is not the first study to use this concept to evaluate ET estimates. In addition, we went further to identify the dominant sources of uncertainties when using this method which is a novelty compared to most other studies that have used this method for evaluating ET estimates. However, inherent uncertainties in the data cannot be a basis for disqualifying the use of this method as hydrologists have to deal with the challenge of uncertainty in data in every study.

---

## Author Comment (AC10)

**Supplementary 1:** Precipitation Estimates from CHIRPS, GPM, PERSIANN-CDR, and their ensemble mean used in the study

[Figure]

**Supplementary 2:** GRACE estimates from CSR, JPL, GSFC and their ensemble mean used in the study

---

## Author Response (AR1)

Dear Editor-in-Chief,

We wish to thank you once again for offering us another chance to revise our manuscript (**hess-2022-185**). We detail below all of the revisions that we have undertaken in response to your recommendations.

With kind regards

Elias Nkiaka (on behalf of the co-authors).

Both reviewers were satisfied with the revisions we made earlier! The Editor also appreciated our detailed and constructive responses to the referees' comments and Editors' decision is publish subject to revisions.

**Response to Editors' comments**

**Comment:** In your revision, please discuss carefully about the uncertainty underlying the data products (precipitation, ET and TWS) and also the used model parameterisations.
**Response:** Thanks for highlighting this issue. In the methods section, we highlighted the uncertainty estimates in precipitation products L284 – 287 and L287 – 290 for TWS (GRACE).
For evapotranspiration estimates, we feel that evaluating the performance of the different products with respect to $ET_{WB}$ using different error metrics (bias and RMSE) provide sufficient information on the uncertainties inherent in the different products.

As for model parameterisations, we acknowledged in various sections of the manuscript that model parameterisation has an important influence on the overall performance of the model. See L92 – 96 and L496 – 500.
However, we feel that investigating the parameterisations schemes underpinning the different hydrological and evapotranspiration models was not one of the objectives of this study. As such, we wish to decline to comment further on this specific issue raised by the Editor.

Comment: I also agree with Reviewer 2 remark on changing the terminology from "ungauged" to "poorly gauged". This is because you have some datasets on discharge in your basins and that can qualify as "poorly gauged" catchments, rather than totally gauged.
**Response:** Thanks very much for insisting on this point. We have now replaced "ungauged" with "poorly gauged" throughout the manuscript. A total of 15 replacements were made.

---

## Author Response (AR2)

Dear Editor-in-Chief,

We wish to thank you once again for offering us another chance to revise our manuscript (**hess-2022-185**). We detail below all of the revisions that we have undertaken in response to the issues raised by reviewer 1.

With kind regards

Elias Nkiaka (on behalf of the co-authors).

Reviewer 2 recommended that the manuscript should be published in its present form!

Reviewer 1 appreciated the fact that we were able to address most of the issues s/he raised during the first round of review. However, this reviewer was still unsatisfied with some aspects of the manuscript and requested additional work to deal with uncertainties in precipitation and GRACE-derived TWSC estimates.
The Editor also appreciated our detailed and constructive responses to the referees' comments but also insisted that we follow the recommendations of reviewer 1 highlighted above.

**Response to reviewer 1 comments**

**Comment:** I have now carefully reviewed the revised version of the manuscript by Nkiaka et al. I find that the authors have addressed many of the concerns that I raised during the first round. However, I still have concerns about the water balance-based ET estimates being used as the 'ground truth'. I elaborate on my concerns below
The authors use only one satellite-based data product for precipitation (CHIRPS) and TWS (GRACE). The uncertainty estimates accounted for in the study are only random errors, which are pertaining to measurement errors from the satellites and not the systematic error or uncertainty, which arises from whether the precipitation or TWS estimate is close to the truth or not. The authors point to other studies which have used the same methodology. However, the use of water balance as a methodology is not the issue here. The way it is used (only one data product per water balance component is). For example, Weerasinghe et al. 2020 (a paper which the authors refer to) uses the average of 3 precipitation datasets over large watersheds. Moreover, most of these studies use these datasets at an annual timescale which averages out sub-annual fluctuations and potentially reduces uncertainty.

**Response:** Thanks for insisting that we provide a more transparent and scientifically robust methodology on how we dealt with uncertainty in the precipitation data used in estimating basin-wide evapotranspiration in our study.

Uncertainty in precipitation input
To deal with uncertainty in precipitation estimates, we used satellite-based precipitation estimates from three different sources including CHIRPS, GPM and PERSIANN-CDR which have been validated and used in other studies in the region. The precipitation products have spatial resolutions of 0.05°, 0.1° and 0.25° for CHIRPS, GPM and PERSIANN-CDR respectively. We then calculated the ensemble mean of the three precipitation estimates for each basin and used this ensemble mean as the precipitation input in this study. See revised manuscript L177 – 187.

**Comment:** Regarding the use of GRACE for small watersheds: Again, the fact that other studies use GRACE for small watersheds in a specific region does not imply that the dataset is suitable over all regions. Here again, the authors misinterpret instrument error as the error/uncertainty in how well GRACE represents TWS. The authors need to justify the use of GRACE over small watersheds better.

**Response:** Thanks for insisting that we provide a more transparent and scientifically robust methodology on how we dealt with uncertainty GRACE-derived TWSC used in estimating basin-wide evapotranspiration in our study.

Uncertainty in GRACE estimates
To minimize errors and uncertainty in the GRACE-derived TWSC estimates, we used an ensemble mean of three GRACE mascon solutions derived from different processing centres including Jet Propulsion Laboratory (JPL) RL06M Version 2.0 GRACE mascon solution with a spatial resolution of 0.5° x 0.5°, Center for Space Research at University of Texas, Austin (CSR GRACE/GRACE-FO RL06 v02 Mascon Grids) with a spatial resolution of 0.25° x 0.25° and NASA GSFC GRACE and GRACE-FO MASCON RL06 v1.0 with spatial resolution of 0.5° x 0.5°. GRACE data were used to estimate basin-wide water balance evapotranspiration ($ET_{WB}$). See revised manuscript L189 – 203.

To further minimize errors and uncertainty in the GRACE-derived TWSC in our smaller-size basins, we re-gridded the GRACE mascon solutions from JPL and NASA to a spatial resolution of 0.25° which is the same spatial resolution for the mascon solutions from CSR. We then proceeded to extract and average the timeseries of all coincident GRACE grid cells for each basin from the three different mascon solutions with the same spatial resolution. Gaps in the time series were infilled using the linear function in Python. Finally, we calculated the ensemble mean of the three solutions to represent GRACE-derived TWSC estimate for each basin. See revised manuscript L264 – 271.

Due to the changes, we made in the methodology, we had to remove sub-section "2.6 Estimating relative uncertainty in basin-scale water balance ET ($ET_{WB}$)" and sub-section "3.2.4 Estimating relative uncertainty in $ET_{WB}$". We also removed Figure 9 in the previous version of the manuscript.

In addition, due to the changes, we made in the precipitation and GRACE-derived estimates used in calculating basin-wide water balance estimates, we changed Figures 6, 7, & 8 and the adjusted the relevant portions of the manuscript accordingly.

We equally replaced the Figure in the supplementary material with two figures showing precipitation and GRACE estimates and their ensemble mean used in our subsequent calculations in the manuscript.